# Dexterous electrical-driven soft robots with reconfigurable chiral-lattice foot design

**Dong Wang** [1,2,3] ✉, **Baowen Zhao**[1,3], **Xinlei Li**[1], **Le Dong**[1], **Mengjie Zhang**[1], **Jiang Zou**[1] **& Guoying Gu** [1,2] ✉

Dexterous locomotion, such as immediate direction change during fast movement or shape reconfiguration to perform diverse tasks, are essential animal survival strategies which have not been achieved in existing soft robots. Here, we present a kind of small-scale dexterous soft robot, consisting of an active dielectric elastomer artificial muscle and reconfigurable chiral-lattice foot, that enables immediate and reversible forward, backward and circular direction changes during fast movement under single voltage input. Our electric-driven soft robot with the structural design can be combined with smart materials to realize multimodal functions via shape reconfigurations under the external stimulus. We experimentally demonstrate that our dexterous soft robots can reach arbitrary points in a plane, form complex trajectories, or lower the height to pass through a narrow tunnel. The proposed structural design and shape reconfigurability may pave the way for next-generation autonomous soft robots with dexterous locomotion.

Biological organisms harness dexterous locomotion as one of their main survival strategies to hunt for food or flee from predators, such as immediate direction change during fast movement or shape reconfiguration to perform diverse tasks. For instance, rabbits run in zig-zag patterns to baffle its follower; the soft bodies of octopuses can alter their shapes, enabling them to squeeze through tiny gaps[1]. By controlling large-scale machines and high-power engines, traditional technology can change direction during high-speed motions. However, the design of dexterous small-scale soft robots is challenging because of the difficulties in the miniaturization of motors and transmission systems and the use of rigid components. Combining structure design and soft smart materials provides alternative possibilities to build miniaturized dexterous soft robots[2–6]. Significant efforts have recently been made to develop soft robots with fast locomotion speed and large output force[7–13]. However, achieving locomotion direction change or shape reconfiguration remains challenging as dexterous locomotion generally requires variations in actuations or structures.

Controlling the actuations sequences is a typical method to realize dexterous locomotion. Examples include a soft robot that forms an "8" path using three dielectric elastomer actuators[14] or a soft-legged quadruped robot selecting different gaits using a pneumatic circuit[15]. However, integrating multiple actuations leads to a large size. The complex actuation sequences and viscous property of the pneumatic flow also hinder high-speed motion and fast direction change[16]. Structural reconfiguration is another way to realize multimodal functions, often generated by multistable transition or material property change in response to external stimuli[17,18]. Examples include soft robots that can transform from ground to air vehicles via the phase change under a thermal stimulus[19], soft robots with origami wheels to pass over gaps by changing the wheel's diameters[20] and direction-changeable soft robots by shape change in inflation and deflation[21]. This method uses structural intelligence to realize multiple functions, whereas it generally suffers from the disadvantages of low response and irreversible motion[22,23].

Moreover, functional materials such as magnetic-responsive soft robots can realize multimodal movement by changing the magnitude and direction of the applied magnetic actuation[24–26]. Still, the workspace is limited within the magnetic field area. Additional functions have also been used, such as the electroadhesive effect[27,28], which is substrate-sensitive and has complex control. Metamaterials are

[1]State Key Laboratory of Mechanical System and Vibration, School of Mechanical Engineering, Shanghai Jiao Tong University, 200240 Shanghai, China. [2]Meta Robotics Institute, Shanghai Jiao Tong University, 200240 Shanghai, China. [3]These authors contributed equally: Dong Wang, Baowen Zhao. ✉ e-mail: wang_dong@sjtu.edu.cn; guguoying@sjtu.edu.cn

designed to show superior properties by structure rather than composition. The structural intelligence of metamaterials has shown attractive promise in designing soft robots with flexible deformation and motion[29]. For instance, Cui et al. developed 3D-architected robotic metamaterials that actively sense and move[30]. Chi et al. proposed a bioinspired multistable architecture that enables a soft swimming robot with high maneuverability and power efficiency[31].

Here we develop a chiral-lattice design method for small-scale soft robots to realize dexterous locomotion. The soft robot consists of a dielectric elastomer artificial muscle as its deformable body and a pair of asymmetric feet: a flat foot and a chiral-lattice foot (Fig. 1). Different from the actuation sequences or structural reconfiguration methods, the multimodal behaviors are benefit from the dynamic resonant and chiral twisting effects. These effects are intrinsically embedded in the lattice structural design, which generates immediate locomotion direction change using a simple input. We show that our developed soft robots exhibit forward, backward and circular motion changes by simply adjusting the voltage frequencies, enabling them to easily reach arbitrary points on a plane by a single actuation: (i) at a low voltage frequency, the flat foot jumps above the ground while the chiral-lattice foot remains on the ground and the soft robot moves right (Fig. 1a); (ii) as the voltage frequency increases, the lattice foot bounce above while the flat foot sticks on the ground and the soft robot moves left (Fig. 1b); (iii) as the frequency increase further, the soft robot moves circularly due to the dynamic resonant and chiral twisting effects (Fig. 1c).

Moreover, the structural design method is combined with the shape memory effect of smart materials to fulfill complex functions, such as forming S-shaped trajectories (Fig. 1d) or passing narrow tunnels. We develop a general analytical, numerical, and experimental framework to predict and program the locomotion direction change and multimodal functions of the soft robot. In this manner, we can program the threshold frequency to control the locomotion direction of soft robots. The experimental results demonstrate that our soft robots can achieve a maximum backward speed of -124 mm s⁻¹, a forward speed of -112 mm s⁻¹ and an angular velocity of around 0.37 rad s⁻¹. The structural design method and shape reconfigurability pave the way for next-generation autonomous soft robots with dexterous locomotion.

## Results

### Structural design

We design an electric-driven small-scale soft robot with dexterous and multimodal motions. The soft robot consists of a dielectric elastomer artificial muscle as its deformable body and a pair of asymmetric feet: a flat foot and a chiral-lattice foot. The dielectric elastomer artificial muscle comprises a pre-stretched dielectric elastomer membrane and a flexible acrylic frame. The dielectric elastomer membrane is biaxially pre-stretched 3.5 × 3.5 times, sandwiched by two compliant carbon grease electrodes, and adhered to the acrylic frame. After relaxing the stretched membrane, the dielectric elastomer artificial muscle buckled into a saddle-shaped structure. Sinusoidal voltage is applied to the electrodes. As the applied voltage on the dielectric elastomer artificial muscle increases or decreases, the body of the soft robot extends or contracts, respectively.

The feet are fabricated by a commercial 3D printer (Stratasys J750) using the material Vero (Supplementary Note 1). The structures of the

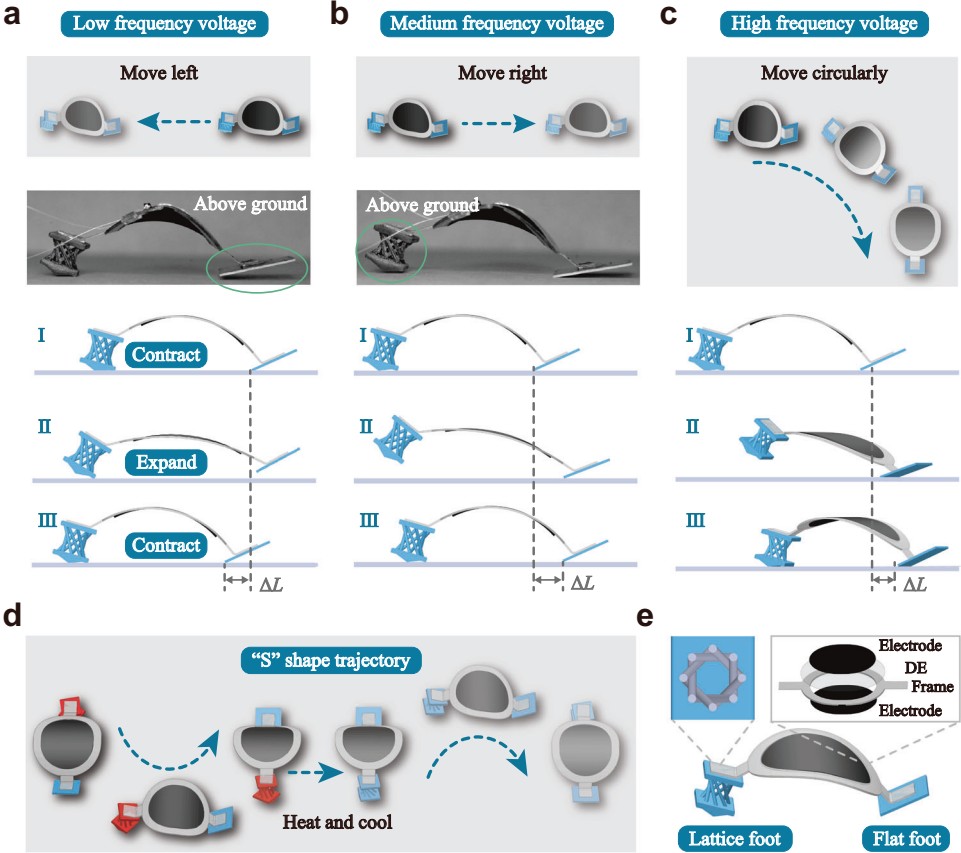

**Fig. 1 | Dexterous soft robots. a** The soft robot moves in the right direction under a low-frequency voltage as the flat foot jumps above the ground. **b** The soft robot moves in the left direction under a medium-frequency voltage as the chiral-lattice foot bounces above the ground due to dynamic behaviors. **c** The soft robot moves in a circular direction at a high-frequency voltage due to resonant and chiral twisting effects. **d** The shape memory effect is combined with the chiral-lattice design method to realize complex functions such as forming an S-shaped trajectory. **e** The soft robot consists of a dielectric elastomer artificial muscle as its deformable body and a pair of asymmetric feet: a flat foot and a chiral-lattice foot.

chiral lattices are shown in Supplementary Fig. 1. The lattices are formed by connecting 1D (straight), 2D (horseshoe), or 3D (spiral) microstructures with the top and bottom plates. The top plate is rotated by an angle $\gamma$ counter-clockwise to form a chiral structure. A twisting can therefore be induced under compression, which is the so-called chiral twisting effect (Supplementary Fig. 1d, e)[32]. The mechanical properties of the chiral lattice can then be tuned by varying the geometric parameters.

The pair of asymmetric feet can generate directional locomotion of the soft robot resulting from friction differences. The multimodal behaviors of the chiral-lattice foot enable the direction change. When a voltage with a low frequency is applied, the lattice foot remains on the ground, and the lighter flat foot jumps above the ground at contraction. Therefore, the flat foot is pulled towards the lattice foot, and the soft robot moves toward the lattice foot direction. In contrast, the lattice foot jumps above the ground under a medium-frequency voltage. The chiral-lattice foot is pulled towards the flat foot, and the soft robot's locomotion direction reverses. When the frequency approaches the natural frequency of the lattice foot, a large compression is

generated due to the dynamic resonant, leading to chiral twisting. The chiral twisting then drives the soft robot to move circularly.

## Dynamics model

To program the dexterous locomotion of the soft robot, we develop a dynamic model to study the multimodal behaviors of the chiral-lattice foot inspired by the ball-bouncing behavior on a substrate[33,34]. The dynamic behaviors of the chiral-lattice foot bouncing on a substrate under a vertical and periodic normal force $F_z$ with a frequency $f$ are studied. $F_z$ is the force component along the vertical direction of the lattice foot generated by the dielectric elastomer artificial muscle (Fig. 2a). The lattice foot is modeled by two rigid plates connected with a spring and a dashpot. The masses of the two plates are the same and denoted by $m$. The spring constant is $k$, and the dashpot coefficient is $\eta$. The vertical upward direction is chosen as the positive $z$-direction.

The following assumptions are made. The contact between the bottom plate and the substrate is rigid; and the inclined angle of the lattice foot $\theta$ remains a constant and only the $z$-direction component of the force generated by the actuator is considered. The force

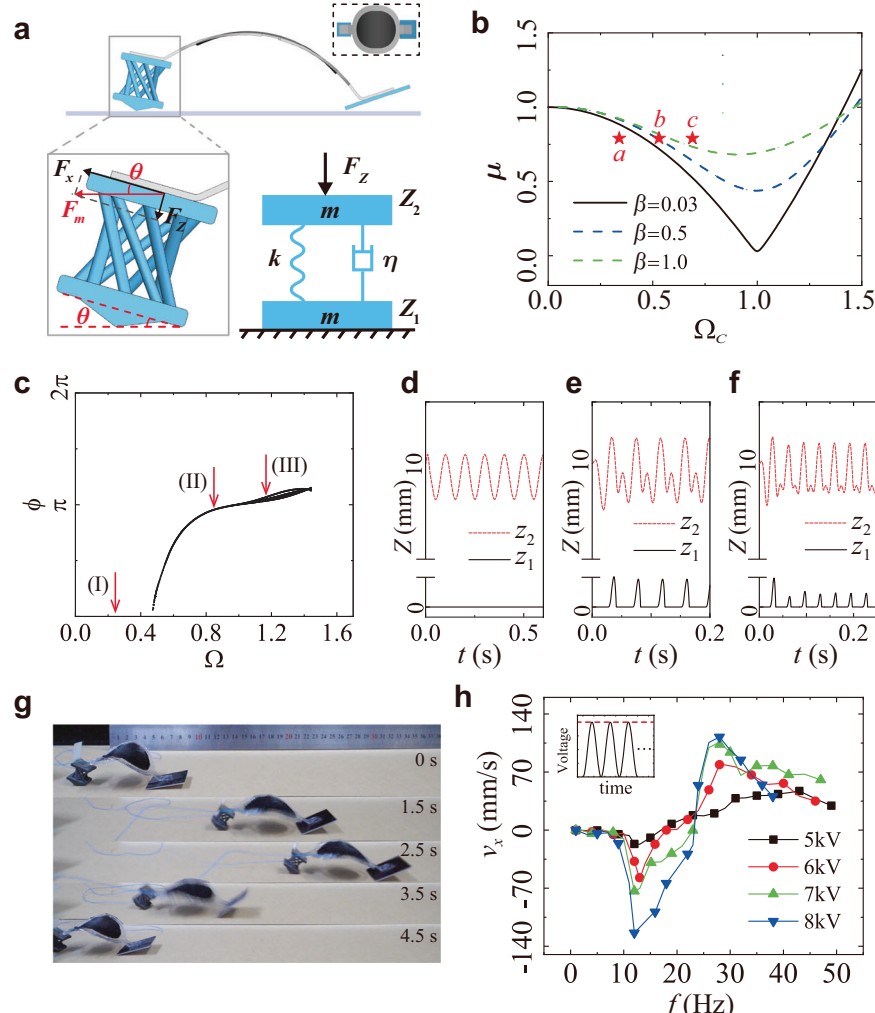

**Fig. 2 | Theoretical modeling and experimental validation of the multimodal locomotion. a** A dynamic model is developed to study the bouncing behavior of the chiral-lattice foot. **b** The theoretically predicted relationship between the normalized force $\mu$ and the normalized critical frequency $\Omega_c$ at different normalized viscosities $\beta = 0.03$, 0.5 and 1.0. Markers $a$, $b$, and $c$ represent the experimental values under a 6 kV voltage at 15 Hz, 23 Hz and 30 Hz, respectively. Markers $a$, $b$ and $c$ fall separately in the no bouncing, critical bouncing and bouncing regions. **c** The bouncing diagram of the lattice foot at $\beta = 0.5$. The dimensionless flight time $\phi$ is

shown as a function of $\Omega$. **d**–**f** Three spatio-temporal diagrams showing the motions of the top (red) and bottom plates (black). The bottom plate stays on the ground at I and bounces at II and III. **g** Experimental snapshots of the immediate locomotion direction change. A 30 Hz voltage is applied first ($t = 0$ to 2.5 s), and a 15 Hz voltage is applied next ($t = 2.5$–4.5 s). **h** Experimental measured $x$-directional speed as a function of applied voltage frequency. Four voltage amplitudes are used: 5, 6, 7, and 8 kV. Moving toward the flat foot is set as the positive direction. The soft robots change locomotion directions at ~23 Hz for all four voltages.

equilibrium in the $z$-direction is written in the following equations:

$$m\ddot{z}_1 = -mg + k(z_2 - z_1 - L) + \eta(\dot{z}_2 - \dot{z}_1) + r, \tag{1}$$

$$m\ddot{z}_2 = -mg - k(z_2 - z_1 - L) - \eta(\dot{z}_2 - \dot{z}_1) + F_z(t), \tag{2}$$

where $z_1$ and $z_2$ represent the positions of the two plates, $g$ is the gravitational acceleration, $r$ is the reaction force between the bottom plate and the substrate. $r$ equals zero when the bottom plate is not in contact with the substrate. $L$ is the initial distance between the two plates. The periodic force $F_z$ can be written in a cosine form $F_z(t) = F_z \cos \omega t$, where $\omega = 2\pi f$. We write the system in dimensionless variables using the following scale changes: $\tilde{t} = \omega t, \tilde{z}_i = z_i m\omega^2 / F_z, \tilde{L} = Lm\omega^2 / F_z$. We now compute the minimal frequency that allows the detachment of the lattice foot from the substrate. The lattice foot is stuck before the detachment indicating $z_1 = 0$. By dropping the tilde, we get

$$\ddot{z}_2 + \frac{\beta}{\Omega}\dot{z}_2 + \frac{1}{\Omega^2}z_2 = \cos t - \frac{1}{2\mu} + \frac{L}{\Omega^2}, \tag{3}$$

$$\frac{\beta}{\Omega}\dot{z}_2 + \frac{1}{\Omega^2}z_2 = \frac{1}{2\mu} + \frac{L}{\Omega^2} - \frac{r}{F_z}. \tag{4}$$

The dimensionless parameters are the normalized force $\mu = F_z / 2mg$ and $\beta = \eta / m\omega_0$ measuring the dissipation between two plates. The dimensionless frequency is $\Omega = \omega / \omega_0$, where $\omega_0 = \sqrt{k/m}$. By solving Eq. (3) with $z_1 = 0$, we can obtain the periodic solution of $z_2$ as

$$z_2 = -\frac{\Omega^2}{2\mu} + L + \frac{\Omega^2}{\sqrt{\beta^2\Omega^2 + (1 - \Omega^2)^2}}\cos(t + \varphi), \tag{5}$$

where $tan\varphi = \frac{\beta\Omega}{\Omega^2 - 1}$. The lattice foot jumps above the substrate as the reaction $r$ becomes zero. By inserting Eqs. (5) into (4) and setting $r = 0$, we can get

$$\sqrt{\frac{1 + \beta^2\Omega^2}{(1 - \Omega^2)^2 + \beta^2\Omega^2}}\cos(t + \varphi + \varphi') = \frac{1}{\mu}, \tag{6}$$

where $tan\varphi' = \beta\Omega$. Equation (6) defines the critical frequency $\Omega_c$ when $\cos(t + \varphi + \varphi') = 1$, which is implicitly expressed as

$$\mu = \sqrt{\frac{(1 - \Omega_c^2)^2 + \beta^2\Omega_c^2}{1 + \beta^2\Omega_c^2}}. \tag{7}$$

The relationship between the applied force and critical frequency $\Omega_c$ separates the regions where the lattice foot remains on the substrate and bounces above the substrate. Figure 2b plots the relationship between $\mu$ and $\Omega_c$ at several viscosities ($\beta = 0.03, 0.5$ and $1.0$). The curve separates the space into two regions. The region below the curve indicates that the lattice foot stays on the ground. The region above the curve shows that the lattice foot bounces above the ground. The soft robot stays stationary for points lying on the curve.

## Forward and backward motion

The model is then used to design the multimodal soft robots. We first measure the force generated by the dielectric elastomer artificial muscle $F_m$ (Supplementary Fig. 7). The generated forces are 0.067, 0.094, and 0.127 N under 5, 6, and 7 kV, respectively. A triangular prism structure is used at the bottom of the lattice foot to guide the

locomotion direction. The angle between the chiral lattice's top surface and the horizontal direction $\theta = 4.8°$. Therefore, the applied force perpendicular to the top surface of the chiral foot can be estimated as $F_z = F_m \times \sin\theta$. When the applied voltage is 6 kV, the corresponding $\mu$ is calculated as 0.79 using $\mu = F_z / 2mg$. The critical $\Omega_c$ is calculated by Eq. (7) as 0.46, 0.47 and 0.50 for $\beta = 0.03, 0.5$ and 1.0, indicating that the effect of $\beta$ can be neglected. The inclined angle of the designed chiral-lattice foot is $\gamma = 90°$. The effective elastic constant $k = 38$ N m$^{-1}$ based on the finite element simulation and experimental results (Supplementary Fig. 5). The mass of the chiral foot is measured as $m = 0.51$ g. The natural frequency of the chiral lattice is then calculated as $f_0 = 2\pi\omega_0 = 43.5 Hz$. The critical frequency is calculated as $f_0 \times \Omega_c = -20\text{--}23$ Hz when $\beta$ ranges from 0.03 to 1.0.

The dynamic behaviors of the chiral-lattice foot can then be obtained using the above physical parameters through the theoretical model. The differential Eqs. (1) and (2) are solved numerically under two different situations: the lattice foot remains on the ground, and the lattice foot jumps above the ground. The dynamic behavior of the lattice foot on the ground is calculated with the condition $z_1 = 0$. At the critical frequency, the lattice foot jumps above the ground, and $r$ becomes 0 while $z_1$ remains 0. The dynamics of the lattice foot above the ground can be solved with the condition $r = 0$, as there is no contact with the substrate. The complete dynamic behaviors can be obtained by solving Eqs. (1) and (2) using different boundary conditions iteratively. A typical diagram for the bouncing lattice foot is plotted in Fig. 2c. The dimensionless flight time $\phi$ is shown as a function of $\Omega$. $\phi = \omega\Delta t$, where $\Delta t$ represents the time between the jumping and landing of the lattice foot. The parameters used are $\mu = 0.79$ and $\beta = 0.5$. The plot allows us to identify bouncing modes. The theoretically predicted $z_1$ and $z_2$ are plotted as a function of time for $\Omega = 0.23$ (10 Hz), 0.81 (35 Hz) and 1.15 (50 Hz) in Fig. 2d–f. The bouncing behaviors under $\beta = 0.03$ are shown in Supplementary Fig. 11 and Supplementary Movie 1. At point I, one can observe that the bottom plate remains on the ground. At points II and III, the bouncing mode in which one bounce of the bottom plate per two oscillations of the top plates is shown.

Based on the theoretical prediction, the chiral-lattice foot dynamics and the soft robots' locomotion behaviors are experimentally studied (Fig. 2g). We first applied a 30 Hz sinusoidal voltage for the first 2.5 s, and a 15 Hz sinusoidal voltage for the next 2 s. The soft robot's moving direction changes from right to left immediately when the voltage frequency changes from 30 to 15 Hz (Supplementary Movie 2). We then captured the movement of the soft robot with a high-speed camera at frequencies of 15 Hz (Fig. 1a), 23 Hz and 30 Hz (Fig. 1b) (Supplementary Movie 3). The theoretical predicted values for 15 Hz, 23 Hz and 30 Hz are represented by markers $a$, $b$, and $c$ in Fig. 2b, which are below, on and above the critical curve, respectively. Experimental results show that the flat foot jumps above the ground while the chiral-lattice foot remains on it at 15 Hz. In contrast, the flat foot remains on the ground while the chiral-lattice foot jumps above the ground at 30 Hz, causing the locomotion direction change. At the critical frequency of 23 Hz, both feet remain on the ground, and the soft robot stays stationary. These experimentally observed phenomena agree well with the theoretical predictions.

The movement mechanisms at low and moderate frequencies are detailed based on the high-speed camera results. Initially, the soft robot is static. When a voltage is applied, the dielectric elastomer artificial muscle expands, and both feet slide in the opposite direction. The soft body contracts when the voltage reduces. As the flat foot is lighter, it jumps while the lattice feet remain on the ground at contraction. Therefore, the flat foot is pulled towards the lattice foot, and the soft robot moves toward the lattice foot direction. In contrast, the lattice foot jumps above the ground under a medium-frequency voltage. The chiral-lattice foot is pulled towards the flat foot, and the soft robot's locomotion direction reverses.

To quantitatively study the dependence of the locomotion behaviors on the voltage frequency, we measured the velocity of the soft robot at various frequencies under different voltage amplitudes, as shown in Fig. 2h. The frequency changes from 0 to around 50 Hz. Four voltage amplitudes are used: 5, 6, 7 and 8 kV. The displacement in the $x$-direction is measured by a high-precision laser displacement sensor (Keyence LK-H085) (Supplementary Fig. 6). The positive $x$-direction (forward) is set towards the flat foot direction. The following phenomena can be observed. The soft robot moves in the negative direction (backward) at a low frequency, and the backward speed increases with the frequency. The magnitude of the backward speed reaches its maximum (124 mm s$^{-1}$) at around 12 Hz and then decreases with the frequency. When the frequency reaches a critical value $f_c$ (around 23 Hz), the soft robot stays stationary. As the frequency increases further, the soft robot moves in the positive direction (forward). The locomotion speed increases with the frequency and reaches its positive maximum (112 mm s$^{-1}$) at around 28–40 Hz. The moving speed generally increases with the applied voltage. The soft robot's body length (BL) is around 91 mm, indicating a maximum negative speed of 1.36 BL s$^{-1}$ and a maximum positive speed of 1.23 BL s$^{-1}$.

## Circular motion at resonance

The chiral twisting effect is induced at resonance due to the chiral structure. By inserting Eq. (5) into Eq. (4), we obtain

$$r = 2mg - F \sqrt{\frac{1 + \beta^2 \Omega^2}{\left(1 - \Omega^2\right)^2 + \beta^2 \Omega^2}} \cos(t + \varphi + \varphi'). \tag{8}$$

Defining the normalized maximum reaction force

$$\kappa = \frac{\max(r) - 2mg}{F} = \sqrt{\frac{1 + \beta^2 \Omega^2}{\left(1 - \Omega^2\right)^2 + \beta^2 \Omega^2}}. \tag{9}$$

$\kappa$ is plotted as a function of normalized frequency $\Omega$ in Fig. 3a. $\kappa$ increases significantly when $\Omega$ approaches the resonance frequency ($\Omega = 1$). The dynamic behavior of the lattice foot under a 6 kV and 46 Hz voltage is numerically solved (Supplementary Movie 4). The model parameters are $L = 0.01$ m, $g = 9.8$ kg m$^{-2}$, $k = 38$ N m$^{-1}$, $m = 0.51$ g, $\eta = 0.07$, $\beta = 0.5$, $F_m = 0.094$ N and $\mu = 0.79$. The corresponding $\kappa$ at $f = 46$ Hz is marked in Fig. 3a.

The theoretically calculated $\Delta z = L - (z_2 - z_1)$ at the steady state under various $f$ is calculated. The dependence of the maximum $\Delta z$ on $f$ is shown in Fig. 3b. The maximum $\Delta z = 0.43$ mm when $f = 46$ Hz. The chiral-lattice foot shows a twisting when compressed. The dependence of the twisting angle $\alpha$ and the axial displacement $\Delta z$ is analyzed by experiments and the finite element method (Fig. 3c). It can be seen that $\alpha$ depends nonlinearly on $\Delta z$. The twisting angle is around 7° at $\Delta z = 0.43$ mm.

Figure 3d shows the locomotion behavior of a soft robot under a voltage of 46 Hz and 7 kV (Supplementary Movie 5). The applied frequency is near the natural frequency. It can be observed that the soft robot rotates in the right direction circularly due to the chiral-twisting effect (Supplementary Fig. 12c). The circular motion is due to the synergistic effect of the dynamic resonant and chiral twisting effects. In one period, the lattice foot is first compressed by force generated by the dielectric elastomer artificial muscle. A twisting is generated simultaneously due to the chiral twisting effect of the lattice foot, causing the rotation of the triangular prism structure. The rotating prism then pushes the soft robots in the right circular direction as the dielectric elastomer artificial muscle contracts. Note that the dependence of $\kappa$ on $\Omega$ is nonlinear. When the voltage frequency is away from the resonance frequency, the reaction force on the lattice foot is

significantly smaller, leading to negligible twisting; the soft robots remain in a forward or backward linear motion.

## Reaching arbitrary points on a plane

Easy control of the fast locomotion direction change of the soft robot can be realized by simply altering the applied frequencies. Figure 3e–h shows that the soft robot can reach different points on a plane (Supplementary Movie 6). Four points at different quadrants are chosen. According to the position of the destination, walking trajectories are planned. The trajectories compose of straight walking and circular walking curves. In all the experiments, a voltage of 6.3 kV is applied. Voltage with three different frequencies is used. Voltages with low (12 Hz) and moderate (25 Hz) frequencies enable backward (blue dashed curve) and forward (white dashed curve) linear motions, respectively. A high frequency (55 Hz) voltage allows circular motion (orange dashed curve).

Circular and straight walking is used to reach point A. Therefore, a 55 Hz voltage is applied first, followed by a 25 Hz voltage. Point B lies in the second quadrant. Therefore, a 12 Hz voltage is applied to induce the reverse direction locomotion after the circular trajectory (55 Hz) to reach point B. A longer time is used for the 12 Hz voltage, and the soft robot arrives at point C in the third quadrant. To reach point D, the soft robot first moves backward (12 Hz), following circular locomotion (55 Hz). The angular velocity is around 0.37 rad s$^{-1}$ at 55 Hz. The ability of the soft robot to reach arbitrary points in a plane overcomes the limitation of the current soft robot's difficulty in control. The developed locomotion change strategy can be applied further to control the soft robot with visual feedback.

## Navigating a maze

We have conducted experiments with a soft robot to navigate a maze, as shown in Fig. 3i and Supplementary Movie 13. Three frequencies are used: 15, 30, and 45 Hz. The soft robot moves forward and backwards and turns right at 15, 30, and 45 Hz, respectively. Turning left is realized by moving backwards and turning right (30 + 45 Hz). The soft robot turns left first, formed by moving backwards (30 Hz from point 1 to 2) and turning right (45 Hz from point 2 to 3). Several right turns and moving forward motions are performed from points 3 to 7. The soft robot turns left to exit (points 7 to 9), formed by a right turn first and then moving backwards (45 + 15 Hz). Simple programming of frequency can achieve forward, backwards, turning left and turning right.

It can be seen that a sharp left turn is difficult as it is a combination of moving backwards and turning right. We have experimentally measured the radius of right turns under various frequencies in Supplementary Fig. 4 and Supplementary Movie 11. We can observe that the radius decreases with the applied frequency. A sharp turn on the right can be achieved with a high frequency. But high frequency will cause the saddle-shaped dielectric elastomer actuator to expand due to the bandwidth and creep, which undermines the forward and backward movement performances.

## S-shaped trajectory

The proposed frequency control strategies use the structural design to change locomotion modes. Complex functions can be achieved by combining structural design and changing material properties in response to external stimuli. Here we demonstrate that the soft robot can form an S-shaped trajectory or lower its height to pass through a narrow tunnel by combining the frequency control strategies with the shape memory effect of the material. We fabricate the lattice foot using a shape memory polymer Vero[35]. The foot can be programmed into various shapes and recovered to its initial state under the thermal stimulus.

As shown in circular motion, the prism is used to control the locomotion direction. A large compression force induces the twisting of the prism to the left side when the voltage frequency approaches

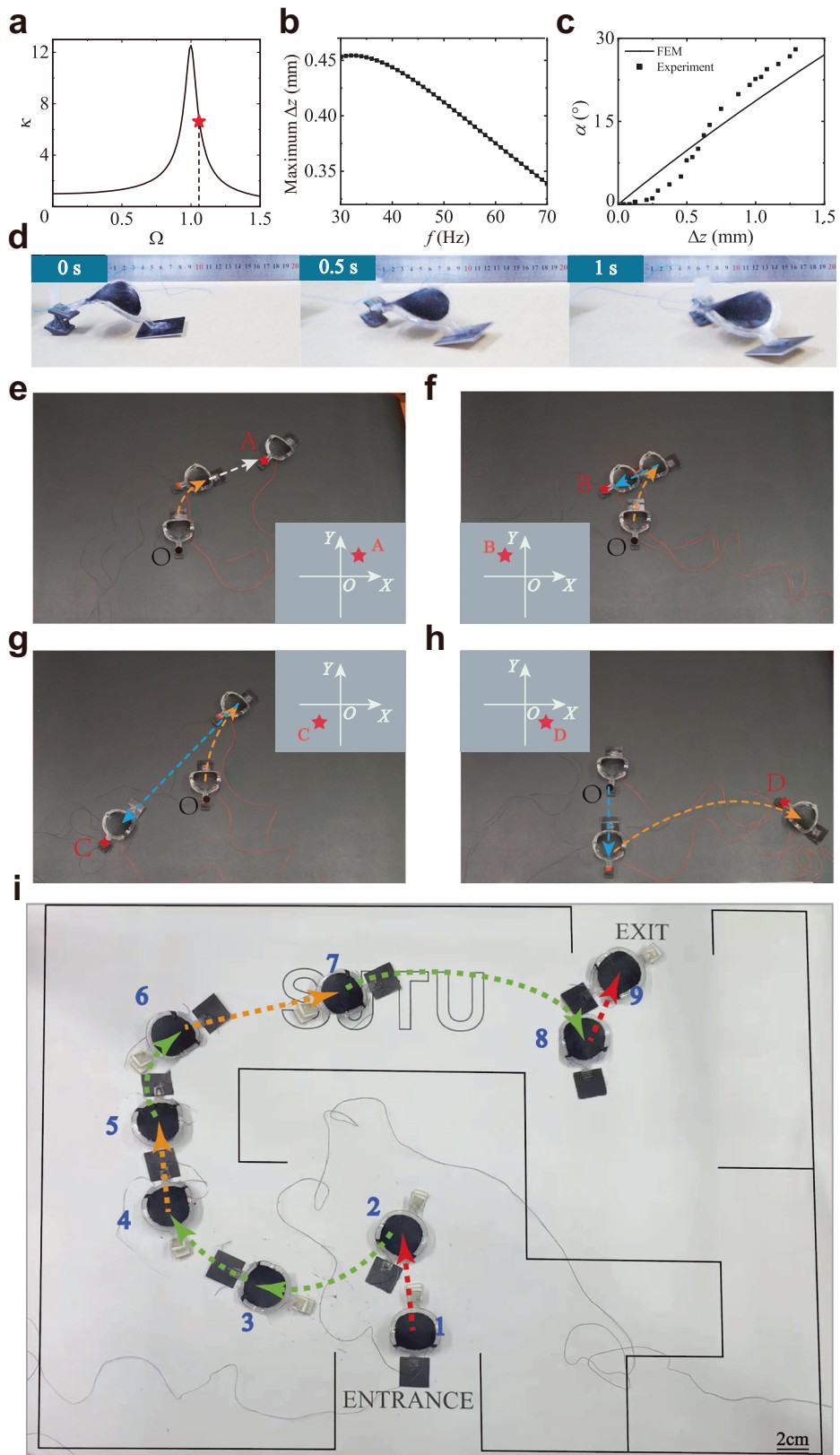

**Fig. 3 | Circular locomotion of the soft robot under a high frequency. a** The theoretically predicted normalized force $\kappa$ as a function of the normalized frequency $\Omega$. The force reaches a maximum at the resonance frequency. **b** The theoretically calculated maximum $\Delta z$ at the steady state under various $f$. **c** The experimental and FE simulated twisting angle $\alpha$ as a function of $\Delta z$. **d** Experimental snapshots of the soft robots moving in the right circular direction. **e**–**h** The soft robot can reach four points A, B, C, and D in the four quadrants by adjusting the frequency only (Supplementary Movie 6). Three frequencies are used to generate backward (12 Hz, blue dashed curve), forward (25 Hz, white dashed curve) and circular (55 Hz, orange dashed curve) motions. **i** Experiments showing a soft robot navigating in a maze.

the resonant frequency, leading to a right-hand side circular motion. Using a similar strategy, both left- and right-hand side motions can be realized by manually adjusting the prism's orientation. Experiments are conducted as shown in Fig. 4. The prism structure is manually oriented with a deviation by an angle of ~30° on the left side (Fig. 4a) and ~10° on the right side (Fig. 4b) with the axial direction. The soft robot then turns to the right or left sides under an applied voltage away from the resonant frequency (Supplementary Fig. 12).

The above strategy is combined with shape reconfiguration under the thermal stimulus to form an S-shaped trajectory that requires both left- and right-hand side circular motions. The shape programming and recovery of the lattice foot are shown in Fig. 4d. The lattice foot is initially twisted, and the prism is oriented with an angle of ~30° to the left at a high temperature ($T_H = 60\,°C$). The deformation is maintained while the temperature is decreased to room temperature ($T_L = 25\,°C$). The loading is removed at $T_L$, and

the lattice is programmed. Under an applied voltage of 40 Hz, the soft robot rotates to the right side. In the recovery step, the lattice is heated back to high temperature, and the original shape of the lattice foot is recovered. The prism is oriented to the right side. The soft robot turns to the left side circularly. The amplitude and frequency of the voltage used are 6.5 kV and 40 Hz.

## Passing through narrow tunnel

A soft robot that can lower its height and pass narrow tunnels is demonstrated (Supplementary Movie 9). A lattice foot consisting of 3D spirals enables a large stretchability. The lattice foot is first programmed to a high position, which can pass environments such as shallow water without affecting the possible onboard electronics. Nevertheless, the height limits its ability to go through narrow tunnels. After applying a thermal stimulus, the lattice foot deforms to the compressed state. The soft robot's height reduces, facilitating the

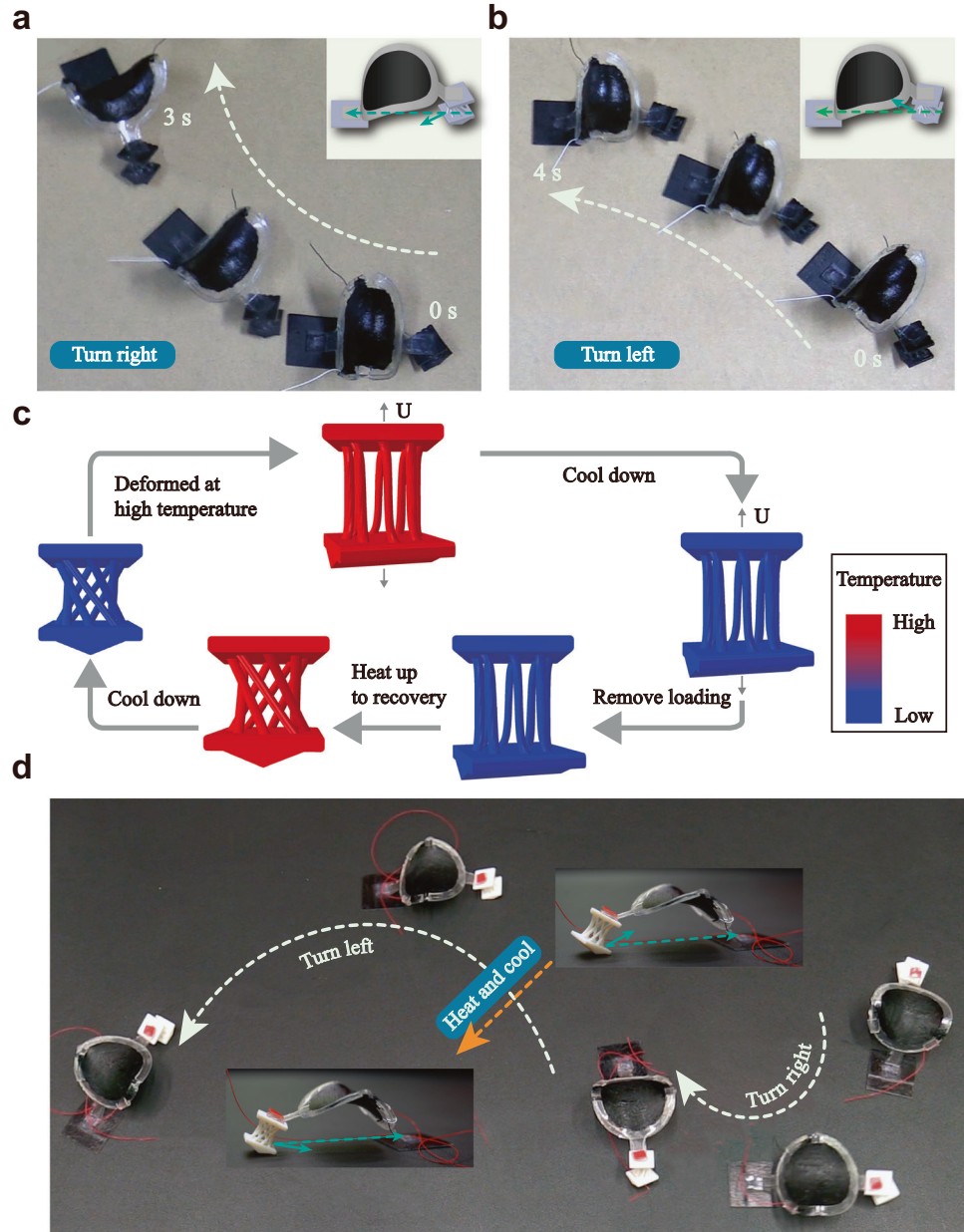

**Fig. 4 | Soft robot forming an S-shaped trajectory under the thermal stimulus.** **a, b** The soft robot moves in the right or left circular directions by orienting the lattice foot to the left or right sides (Supplementary Movie 7). **c** FE simulated shape memory cycles of the lattice foot. **d** An S-shaped trajectory is realized by combining the direction control strategy with the shape memory effect (Supplementary Movie 8).

pass-through of the narrow tunnel. The FE-simulated shape reconfiguration of the lattice foot is shown in Fig. 5a. Figure 5b–e shows the complete locomotion strategies for the soft robot. Initially, the soft robot moves forward with 25 Hz, 6.8 kV and is blocked by the narrow tunnel. The soft robot then moves backward under a 12 Hz voltage. The lattice foot is heated to reduce the soft robot's height from 41.6 mm to 37.2 mm, enabling it to pass through the narrow tunnel. The lattice foot is cooled down to fix the structure. The soft robot can then pass through the narrow tunnel under a 25 Hz voltage and reverse to its original position when a 12 Hz is applied. The soft robot can achieve multiple S shapes by designing the chiral-lattice foot (Supplementary Movie 12). The effects of substrates and scalability on the locomotion of the soft robots are also studied (Supplementary Movies 14–16). The soft robots can also be embedded with heaters (Supplementary Movie 17) and carrying loads (Supplementary Movie 18). Additional experimental results on locomotion and shape memory effect are shown in Supplementary Note 2 and 3. Effects of other types of feet are shown in Supplementary Note 4, Supplementary Fig. 19, and Supplementary Movie 10.

## Discussion

In this work, we report a chiral-lattice design method for soft robots that enables the realization of immediate direction change during fast movement. Forward, backward and circular motions are formed by simply controlling the applied voltage frequency. A saddle-shaped dielectric elastomer artificial muscle drives the soft robot, and a pair of asymmetric feet generates asymmetric motions: a flat foot and a chiral-lattice foot. The soft robot moves backward at low frequency as the lightweight flat foot is pulled above the ground. It moves forward as the lattice foot jumps above the ground under the dynamic force when the frequency exceeds a threshold. As the frequency approaches the natural frequency of the lattice foot, a large reaction force is generated due to the resonance behavior, and the twisting energy is stored attributing to the chiral structure, which synergistically turns the soft robots. We develop a dynamic theoretical model to investigate the bouncing dynamics of the lattice foot. It predicts the threshold separating regions where the lattice foot remains on or jumps above the ground. The soft robots' circular motion is predicted by the dynamic resonance and chiral twisting effects of the lattice foot.

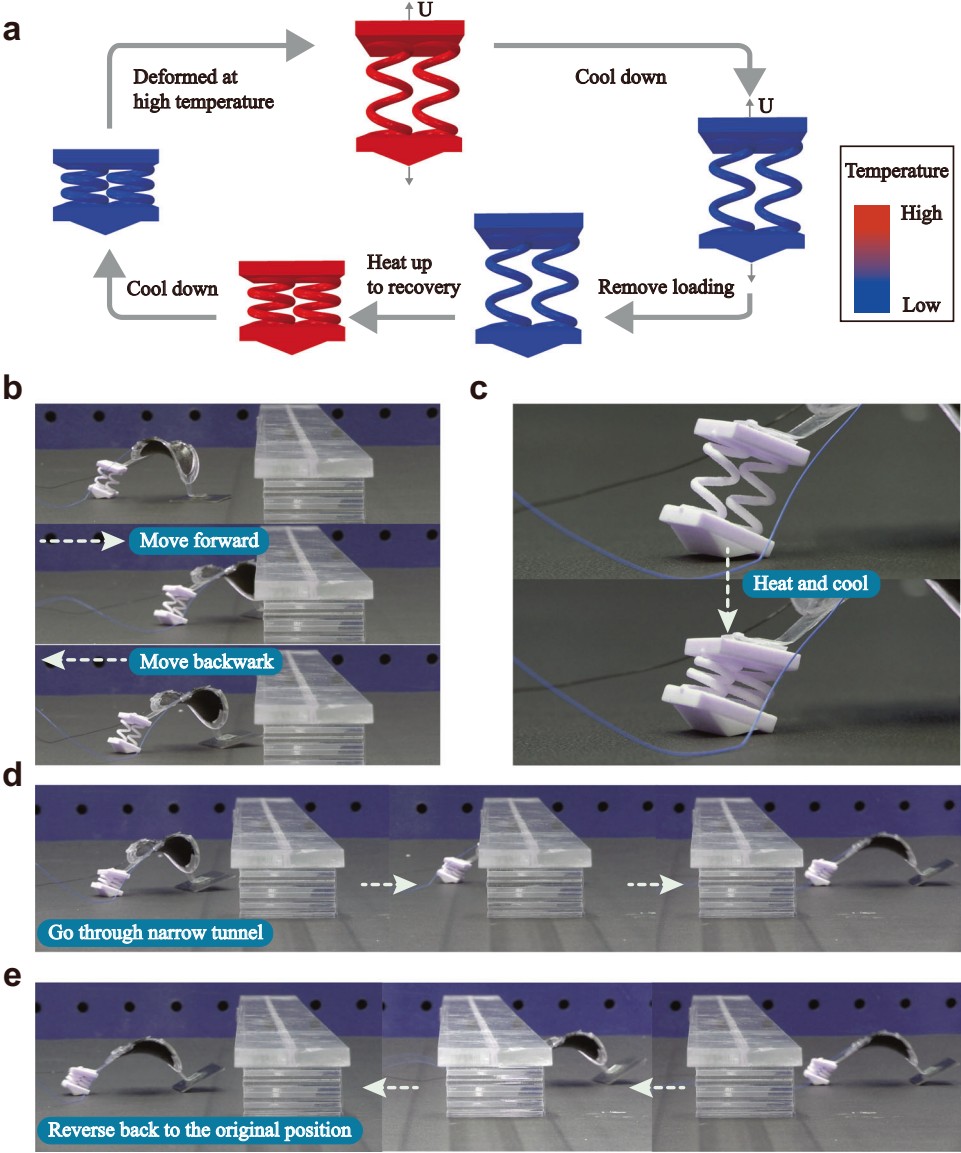

**Fig. 5 | Demonstrations of the soft robot passing through a narrow tunnel.**
**a** The FE simulated shape memory cycle of the multimodal soft robots. **b** The soft robot moves forward first (25 Hz) and then backward (12 Hz). **c** A thermal stimulus is applied to activate the shape reconfiguration. **d** The soft robot can go through a narrow tunnel under a 25 Hz voltage. **e** The soft robot can reverse to its original position under a 12 Hz voltage.

Demonstrations of soft robots reaching points in four different quadrants are realized by changing only the applied frequencies. The ability of direction changes results from the intrinsically dynamic resonant and chiral twisting properties of the lattice structure, which can be further integrated with material properties to fulfill complex functions. The soft robots can follow complex S-shaped trajectories under a thermal stimulus. By replacing the straight rod with a spiral structure in the lattice, we demonstrate a soft robot moving forward, backward, and lower height to pass through the narrow tunnel and return to the original position. The developed soft robots have the ability of dexterous motions by controlling the frequency only, which can support a wide range of applications, such as search and rescue, exploration and inspection, etc. The soft robots can navigate through the unstructured environment and squeeze into tight spaces, making them ideal for search and rescue missions. They can navigate complex environments and gather scientific research or inspection data. The electric-driven mechanism also allows for ready integration with sensors and control circuitry, enabling the design of autonomous soft robotics with autonomous behaviors, such as navigation, obstacle avoidance, and task completion, even under external disturbances. The structural design method and shape reconfigurability pave the way for next-generation autonomous soft robots with dexterous locomotion.

## Methods

### Fabrications

The flat and lattice feet are 3D printed by a commercial printer (Object J750, Stratasys), made of shape memory polymer material Vero. The flat and lattice feet are glued with two ends of the dielectric elastomer artificial muscle to form the soft robot. Dielectric elastomer artificial muscles are used to drive the soft robot because of their high speed, large actuation strain, high power density, and simple materials. The manufacturing procedures of the dielectric elastomer artificial muscle are shown in the Supplementary Fig. 9.

### Blocking forces measurements

One end of the actuator is fixed on a plate, and the other is connected to a force sensor (Transform, K3D40) (Supplementary Fig. 7a). Initially, the soft robot is at the rest state, and the blocking force equals zero. We then apply a sinusoidal voltage and record the force sensor data using a high-voltage amplifier (TREK 10/10C-HS) and a dSPACE-DS1103 board. A 5 kV sinusoidal voltage is applied with a frequency of 20 Hz, as shown in Supplementary Fig. 7b. The corresponding output force is shown in Supplementary Fig. 7c. The Fourier transform of the output force is shown in Supplementary Fig. 7d. It can be seen that the primary peak frequency is 20 Hz, which is the same as the applied voltage. The output forces are also measured under an applied voltage of 6 and 7 kV at different frequencies from 1 to 45 Hz. The average maximum forces are calculated and shown in Supplementary Fig. 7e. The magnitude of the output force increases with the applied voltage and is not sensitive to the frequency. The phase differences between the output forces and applied voltages are measured (Supplementary Fig. 7f). The phase difference increases with the applied frequencies due to the viscoelastic properties of the VHB dielectric material.

### Finite element simulation procedures

The finite element simulations were carried out using the commercial finite element software ABAQUS (3DS Dassault Systèmes, France). The finite element simulation details in the uniaxial compression simulations are described. 3-node quadratic hybrid beam elements (B32H) are used with refined meshes to improve computational accuracy. Quasi-static simulations are performed using Abaqus/Standard. The Vero is modeled as a linear elastic material with Young's modulus $E = 1.5$ GPa (Supplementary Fig. 5e). The geometry of the chiral lattice foot is shown in Supplementary Fig. 2. The lower surface end of the specimen is fixed, and the upper surface is subjected to the displacement loading condition along the vertical direction. We use Python scripts to automatically implement the finite element simulations for specimens with different design parameters.

Finite element simulations are also conducted to analyze the shape memory effects of the designed lattice structures. A multi-branch model developed in our previous study[36] captures the thermo-mechanical behaviors of the chiral-lattice foot. The model consists of one equilibrium branch associated with the elastic response and 30 non-equilibrium branches associated with the viscoelastic response. The non-equilibrium branches are represented by the Maxwell model. Multi-branch model is achieved by implementing the fitted viscoelastic parameters into the time-domain Prony-series model in ABAQUS material library with a UTRS subroutine to describe the time-temperature superposition. One end of the chiral-lattice foot was fixed, and displacement loading was applied to the other. The 10-node modified thermally coupled second-order tetrahedron and hourglass control element C3D10MT is used to perform the calculation. There are five steps in the simulation. In the first step, the object is heated to 60 °C. In the second step, the displacement boundary condition is created, leading to twisting and elongating. In the third step, the displacement boundary condition is fixed, and the temperature is cooled down to 25 °C. In the fourth step, the displacement boundary condition is removed. In the final step, the object is heated to 60 °C and gradually returns to its original shape.

## Data availability

The data that support the findings of this study are available within the article and its Supplementary Information and from the corresponding authors upon reasonable request.

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

## Acknowledgements
This work was supported by the National Natural Science Foundation of China (Grant Nos. 52275025, 52025057, T2294372012), the Science and Technology Commission of Shanghai Municipality (Grant No. 20550712100), and the State Key Laboratory of Mechanical System and Vibration (grant no. MSVZD202212).

## Author contributions
G.G. and D.W. conceived the concept. B.Z., X.L., M.Z., and J.Z. conducted the experiments. D.W., B.Z., and L.D. performed data analysis, simulation, and modeling. D.W. and G.G. wrote the paper.

## Competing interests
The authors declare no competing interests.
