## [Peer Review File · Nature Communications]

Dexterous electrical-driven soft robots with reconfigurable chiral-lattice foot designREVIEWER COMMENTS

Reviewer #1 (Remarks to the Author):

To address the challenge in maneuvering soft robots that often require multiple actuation inputs and complex controls, the authors reported using a single actuation input to the dielectric body alongside chiral lattice feet to achieve different forward, reverse, circular, and S-shaped paths etc. Tuning the actuation frequency induces vibration and twisting in the 3D chiral lattice foot to tune the asymmetric frictions of the feet for achieving different directional motions. This work creatively combines structural intelligence in the metamaterials and soft materials actuation to achieve high-performance robotic functions. This is also a solid work with the designs guided by the developed dynamics models. Overall, this is a very interesting and high-quality work that merits its acceptance for publication in Nat. Comm. Some minor comments are below:

(1) In the introduction part, brief discussions on the strategy of combining structural intelligence of metamaterials and robotics are suggested to add. Related references are: Rafsanjani, et al., "Programming soft robots with flexible mechanical metamaterials", *Sci. Robot.*, 2019, DOI: 10.1126/scirobotics.aav7874; Cui et al., "Design and printing of proprioceptive three-dimensional architected robotic metamaterials", *Science*, 2022, Vol 376, pp. 1287-1293, DOI: 10.1126/science.abn0090; and Chi et al., "Snapping for high-speed and high-efficient butterfly stroke-like soft swimmer", *Sci. Adv.* 2022, 8 (46), eadd3788, DOI: 10.1126/sciadv.add3788. The last two are also closely related to utilize structure designs for maneuverable motion in soft robotics.

(2) Technical question on the correlation between the circular motion behavior and twisting in the chiral lattice. Fig. S1 shows the twisting angle has a range and increases with the applied compression force. During actuation, how large is this twisting angle and how it changes with the compression force from the body under different frequency? Also, how this twisting angle guides the radius of the circular path?

(3) The S-shaped path is interesting by changing the circular direction. How fast can it switch from left to right-hand circular motion at what temperature? After going one circle S, will it be possible to reheat again to back to the original circular? Two circles may be needed to show its controllability.

Reviewer #2 (Remarks to the Author):

The authors have presented a dexterous soft robot with dielectric elastomer actuator (DEA) with asymmetric feet. Although several DEA-based soft actuators and robots have been proposed in the literature based on similar materials and DEA geometry, the reconfigurability of the chiral-lattice foot and the ability to drive the soft robot forward, backward and circular motion with the change in frequency are impressive. Overall, the manuscript is well written and further clarification of the following points will strengthen the paper to be considered for publication in Nature Communications.

Major Comments:

1. The authors have demonstrated the navigation to arbitrary points and through a tunnel. It would be interesting to see how the soft robot can navigate in a narrow path with complex pattern like a maze. Would it be able to take sharp turns at the corners? Would it require complex programming of the frequency?

2. From the supplementary videos of the soft robots, there is noticeable tearing of the DEA at some locations. This is a known challenge for DEAs where repeated use would cause significant wear ultimately leading to failure. What is the break down voltage of the DEA geometry used here? Have the authors performed cyclic tests to determine the repeatability in performance and the lifetime of their soft robot?

3. What is the effect of the substrate on the speed of the soft robot? How does paper compare to silicone, glass to sand paper?

4. How does the speed and performance scale with the size of the soft robot? Is it scalable?

5. Have the authors performed any experiments to incorporate heaters into the design so that the chiral-lattice can be reconfigured without manual input of thermal stimulus?

6. What other applications can the authors envision for the soft robot beyond locomotion? Can it carry a load? What are the effects of a load on the performance? Does a slight change in the load still allow for the lifting of the flat and chiral feet to enable locomotion?

Minor Comments:

1. Minor grammar corrections are needed.

Reviewer #3 (Remarks to the Author):

This paper proposes a novel strategy to control a soft robot driven by DE. The core module in the robot is the so called chiral-lattice foot. Based on the dynamic response of the foot, the robot can move forward, backward and in circular direction via single voltage input. The innovative design method can inspire the research in soft robot and flexible structures. The operation principle is practical and highly valuable for the robotic system in applications. The manuscript is well organized and clearly presented to accept.

Comments in details:

1. More detailed design parameters of the robot should be provided, including both feet and the DEA. Also, the magnitudes of the parameters in Fig. S1 are missed.

2. The diagram in Fig. 2a should be the front view.

3. In page 7, line 190, "0.0671 N, 0.0941 N and 0.127 N", the accuracy is not uniform. In line 197, the author claimed "The effective elastic constant $k = 38 \text{ N/m}$ based on the finite element simulation results (Fig. S3)". The experimental data of "k" should be provided and compared with the simulation results. And more details of the FEA model should be provided, such as the material property and element type. In addition, the mechanical test data of the materials in this work should be provided. In line 188-200, many design parameters and test data are mentioned, it will be helpful to add more details and illustrations.

4. In Fig. S4e,f, the data under 6 kV input is much fewer than that of 5 kV and 7 kV with frequency ranges from 35 Hz-45 Hz. The reason should be presented.

5. In page 8, line 208, the dynamic equations should be Eqs (1) and (2), when the robot bounces off the ground, since z_1 is no longer being zero. In line 224-225, the marker b in Fig. 2b indicates that $\beta=0.5$ in the chiral-lattice system. However, β equals 0.03 in Fig. 2c,d. The parameter should keep the same after calibration.

6. In page 9, line 265-266, the authors claimed that, "The dynamic behavior of the lattice foot under a 7kV and 46 Hz voltage is numerically solved.". The model parameters should be presented. In line 269-270, the authors claimed that, "The dependence of the twisting angle α and the axial displacement Δz is analyzed using the finite element method (Fig. 3(c))." The experimental results should be presented. In line 272, is the chiral-lattice foot left-handed?

7. In page 12, line 330-332, is the "deviation" angle same as γ in Fig. S1? Why different deviation angles lead to different deviation direction of the robot? The relationship between the chirality and deviation angle of the chiral-lattice foot and the deviation direction of the robot should be discussed in detail.

8. In Movie S8, the FEA is presented without details, which should be illustrated in main text or supplementary materials.

Response to Reviewer 1

General comments:

To address the challenge in maneuvering soft robots that often require multiple actuation inputs and complex controls, the authors reported using a single actuation input to the dielectric body alongside chiral lattice feet to achieve different forward, reverse, circular, and S-shaped paths etc. Tuning the actuation frequency induces vibration and twisting in the 3D chiral lattice foot to tune the asymmetric frictions of the feet for achieving different directional motions. This work creatively combine structural intelligence in the metamaterials and soft materials actuation to achieve high-performance robotic functions. This is also a solid work with the designs guided by the developed dynamics models. Overall, this is a very interesting and high-quality work that merits its acceptance for publication in Nat. Comm.

Response to General Comments:

Thank you very much for your time in reviewing our paper and for the insightful and constructive comments. We greatly appreciate your recognition of our work. To address your comments and concerns in full, we have significantly improved our manuscript and added a substantial amount of new data, analyses, discussions, and clarifications (**with revised 2 Figures, newly added 11 Supplementary Figures, and revised 2 Supplementary Figures**). In the following paragraphs, we will address your comments point-by-point. Sentences newly inserted into the manuscript and the supplementary information are marked in **red**.

Specific Comments:

Comment 1:

Some minor comments are below:

*In the introduction part, brief discussions on the strategy of combining structural intelligence of metamaterials and robotics are suggested to add. Related references are: Rafsanjani, et al., "Programming soft robots with flexible mechanical metamaterials", *Sci. Robot.*, 2019, DOI: 10.1126/scirobotics.aav7874; Cui et al., "Design and printing of proprioceptive three-dimensional architected robotic metamaterials", *Science*, 2022, Vol 376, pp. 1287-1293, DOI: 10.1126/science.abn0090; and Chi et al, "Snapping for high-speed and high-efficient butterfly stroke-like soft swimmer", *Sci. Adv.* 2022, 8 (46), eadd3788, DOI: 10.1126/sciadv.add3788. The last two are also closely related to utilize structure designs for maneuverable motion in soft robotics.*

Response to comment 1:

We thank the reviewer for the suggestion. We felt sorry for missing the imported reference in our original manuscript. In the revised manuscript, we have cited them and added a brief discussion, which are reflected in the following sentences.

[On Page 2 of the revised manuscript]

Metamaterials are designed to show superior properties by structure rather than composition. The structural intelligence of metamaterials has shown attractive promise in designing soft robots with flexible deformation and motion ([*Science Robotics* 4.29 (2019): eaav7874]). For instance, Cui et

al. developed 3D-architected robotic metamaterials that actively sense and move ([*Science* 376.6599 (2022): 1287-1293]). Chi et al. proposed a bioinspired multistable architecture, enabling a soft swimming robot with high maneuverability and power efficiency ([*Science Advances* 8.46 (2022): eadd3788]).

Comment 2:

Technical question on the correlation between the circular motion behavior and twisting in the chiral lattice. Fig. S1 shows the twisting angle has a range and increases with the applied compression force. During actuation, how large is this twisting angle and how it changes with the compression force from the body under different frequency? Also, how this twisting angle guides the radius of the circular path?

Response to comment 2:

Thank you for the constructive comments. In the revised manuscript, we have presented the dependence of the largest twisting angle α under different frequencies f (Fig. S4(a)) by combining the experimental and theoretical methods. Fig. S4(d) also shows the measured radius of the circular path under various f . We can see that there is no strong correlation between the twisting angle and the radius of the circular path, as the radius depends on not only the twisting angle but also the forward speed. The forward speed decreases significantly when the frequency increases. Therefore, when a large f is applied, the soft robot twists almost at its original position. To clarify this point, we have added the following paragraph in the revised SI of the manuscript.

[On Page 7 of the revised SI]

The dependence of the largest twisting angle α under different frequencies f is plotted in Fig. S4(a) by combining the experimental and theoretical methods. We first calculate the maximum Δz at the steady state under various f (Fig. S4(b)). Fig. S4(c) compares the FEM simulated and experimental α and Δz . A reasonable agreement is achieved. The relationship between α and f is then obtained using Fig. S4(b) and (c). We can observe that the largest twisting angle decreases slightly from $\sim 7.6^\circ$ to $\sim 4.7^\circ$ when f increases from 30 Hz to 70 Hz.

We measured the rotation curvature of the soft robot as a function of the frequency, as shown in Fig. S4(d). We can observe that the radius decreases significantly when the applied frequency decreases from 40 to 50 Hz. When the applied frequency reaches 50 Hz, the rotation of the soft robot is almost in position. This phenomenon can be explained below. The radius depends on both the twisting angle and forward speed. The forward speed decreases significantly when the frequency increases due to the bandwidth and creep of the DE actuator. Movie S11 shows the turning motion of the soft robot under $f = 54$ Hz. We can observe that the soft robot twists almost at its original position. The saddle shape DE actuator keeps at an expanded state at high frequency, which cannot provide enough actuation force for the forward motion.

Fig. S4(a) The dependence of the largest twisting angle α under different frequencies f . (b) The theoretically calculated maximum Δz at the steady state under various f . (c) FEM simulated and experimental α vs Δz . (d) The dependence of the experimentally measured radius of the circular path on f .

Comment 3:

The S-shaped path is interesting by changing the circular direction. How fast can it switch from left to right-hand circular motion at what temperature? After going one circle S, will it be possible to reheat again to back to the original circular? Two circles may be needed to show its controllability.

Response to comment 3:

Thank you for the kind comments and suggestion. We have measured the switch time, conducted experiments to form multiple circular shapes, and explored a maze to show controllability, as shown below and marked in red in the revised manuscript and SI.

[On Pages 16-18 of the revised SI]

S3.1. Switch time

The switch from left to right-hand circular motion is around 2 mins. The temperature is around 90 $^\circ$ C. The switch time includes both the heating and cooling time. We have experimentally measured the transition time upon heating under various temperatures (Fig. S16). The material of the chiral

lattice foot is a shape memory polymer with a glass transition temperature of ~ 58 °C. The transition time change from ~ 22 s to ~ 1 s when $T = 50$ °C to 95°C. The cooling time is relatively longer. Embedding cooling devices effectively reduces the cooling time, which we will explore as future work.

S3.2. Multiple circles

After going one circle S, it is impossible to reheat again to return to the original circular as the Vero material is a one-way shape memory polymer. The lattice foot can only transform from the programmed state to the original state upon heating, but not the reverse. If a two-way shape memory polymer is used, it is possible to realize multiple S shapes. We can also use structural design to achieve two circles (Movie S12). We design a bilayer lattice foot, as shown in Fig. S17(a). Both layers are programmed first (Fig. S17(b)). Under an electric actuation with $f = 45$ Hz, the soft robot turns right (A to B in Fig. S17(e)). After it reaches point B, a thermal stimulus is applied to the bottom layer of the foot, and it recovers to its original shape (Fig. S17(c)). The triangular prism forms an angle with the centerline of the soft robot due to the twist of the bottom layer, which drives the soft robot to turn left from B to C. We then heat the top layer, and the prism changes to the left direction. The soft robot turns right (C to D). We need to mention that the above design method is unstable due to heat conduction. A stable way to realize a complex trajectory is by applying different frequencies. We have controlled the soft robot to explore a maze by varying the frequency to show its controllability, as shown in Movie S13.

Fig. S16. The transition time t from the programmed state to the original shape at various temperatures T .

Fig. S17. (a) The initial shape, (b) programmed state, (c) recovered state I and (d) recovered state II of a bilayer lattice foot. (e) The moving trajectory of the soft robot.

[On Page 12 of the revised manuscript]

Navigating a maze

We have conducted experiments with a soft robot to navigate a maze, as shown in Fig. 3(i) and Movie S13. Three frequencies are used: 15 Hz, 30 Hz and 45 Hz. The soft robot moves forward and backwards and turns right at 15 Hz, 30 Hz and 45 Hz, respectively. Turning left is realized by moving backwards and turning right (30 Hz + 45 Hz). The soft robot turns left first, formed by moving backwards (30 Hz from point 1 to 2) and turning right (45 Hz from point 2 to 3). Several right turns and moving forward motions are performed from points 3 to 7. The soft robot turns left to exit (points 7 to 9), formed by a right turn first and then moving backwards (45 Hz + 15 Hz). Simple programming of frequency can achieve forward, backwards, turning left and turning right.

It can be seen that a sharp left turn is difficult as it is a combination of moving backwards and turning right. We have experimentally measured the radius of right turns under various frequencies in Fig. S4 and Movie S11. We can observe that the radius decreases with the applied frequency. A sharp turn on the right can be achieved with a high frequency. But high frequency will cause the saddle-shaped dielectric elastomer actuator to expand due to the bandwidth and creep, which undermines the forward and backward movement performances.

Fig. 3(i). Experiments showing a soft robot navigating in a maze.

We greatly appreciate your insightful comments and suggestions again, which have substantially improved our manuscript. We hope our revised manuscript and responses can address your concerns and comments. Thank you

Response to Reviewer 2

General comments:

The authors have presented a dexterous soft robot with dielectric elastomer actuator (DEA) with asymmetric feet. Although several DEA-based soft actuators and robots have been proposed in the literature based on similar materials and DEA geometry, the reconfigurability of the chiral-lattice foot and the ability to drive the soft robot forward, backward and circular motion with the change in frequency are impressive. Overall, the manuscript is well written and further clarification of the following points will strengthen the paper to be considered for publication in Nature Communications.

Response to General Comments:

Thank you very much for your time in reviewing our paper and for the insightful and constructive comments. We greatly appreciate your recognition of our work. To address your comments and concerns in full, we have significantly improved our manuscript and added a substantial amount of new data, analyses, discussions, and clarifications (with revised 2 Figures, newly added 11 Supplementary Figures, and revised 2 Supplementary Figures). In the following paragraphs, we will address your comments point-by-point. Sentences newly inserted into the manuscript and the supplementary information are marked in red.

Specific Comments:

Comment 1:

Major Comments:

The authors have demonstrated the navigation to arbitrary points and through a tunnel. It would be interesting to see how the soft robot can navigate in a narrow path with complex pattern like a maze. Would it be able to take sharp turns at the corners? Would it require complex programming of the frequency?

Response to comment 1:

We thank the reviewer for the constructive comments. Based on your suggestion, we have added new set of experiments with our soft robot to navigate a maze. We can see that by programming of the frequency, our soft robot can achieve forward, backwards, turning left and turning right. We have added the following paragraphs in the revised manuscript and SI to clarify these points.

[On Page 12 of the revised manuscript]

Navigating a maze

We have conducted experiments with a soft robot to navigate a maze, as shown in Fig. 3(i) and Movie S13. Three frequencies are used: 15 Hz, 30 Hz and 45 Hz. The soft robot moves forward and backwards and turns right at 15 Hz, 30 Hz and 45 Hz, respectively. Turning left is realized by moving backwards and turning right (30 Hz + 45 Hz). The soft robot turns left first, formed by moving backwards (30 Hz from point 1 to 2) and turning right (45 Hz from point 2 to 3). Several

right turns and moving forward motions are performed from points 3 to 7. The soft robot turns left to exit (points 7 to 9), formed by a right turn first and then moving backwards (45 Hz +15 Hz). Simple programming of frequency can achieve forward, backwards, turning left and turning right.

It can be seen that a sharp left turn is difficult as it is a combination of moving backwards and turning right. We have experimentally measured the radius of right turns under various frequencies in Fig. S4 and Movie S11. We can observe that the radius decreases with the applied frequency. A sharp turn on the right can be achieved with a high frequency. But high frequency will cause the saddle-shaped dielectric elastomer actuator to expand due to the bandwidth and creep, which undermines the forward and backward movement performances.

Fig. 3(i). Experiments showing a soft robot navigating in a maze.

[On Page 7 of the revised SI]

Fig. S4(d). The dependence of radius on the applied frequency.

Comment 2:

From the supplementary videos of the soft robots, there is noticeable tearing of the DEA at some locations. This is a known challenge for DEAs where repeated use would cause significant wear ultimately leading to failure. What is the break down voltage of the DEA geometry used here? Have the authors performed cyclic tests to determine the repeatability in performance and the lifetime of their soft robot?

Response to comment 2:

Thank you for the constructive comments. We have added the referred information and performed new experiments to evaluate the repeatability and lifetime of the soft robot. To clarify these points, we have added the following sentences in the revised manuscript and SI..

[On Page 9 of the revised SI]

S2.3 Cyclic tests

The breakdown voltage is around 8 kV. When the voltage exceeds this value, the DEA may break down by the high voltage and burn. The breakdown voltage depends on the deformed thickness of the DEA membrane. The thickness of the DEA is 1 mm, and the deformed area is 3.5×3.5 times the original area.

We have conducted cyclic tests of the saddled-shaped DE soft actuator using self-built equipment (Fig. S8(a)). The left side of the soft actuator is attached to a platform driven by a motor. A cyclic displacement of ± 3 mm is applied on the platform. The applied frequency is 30 Hz. The right side of the soft actuator is attached to a force sensor. Positive or negative force shows that the soft actuator is in expanded or contracted states. The average maximum and minimum forces during one minute are extracted for 6 hours with a 30 mins interval (Fig. S8 (b)). Fig. S8(c) shows the measured force curves during one minute at the start, 2 hours later and 4 hours later. From the experimental results, we can observe that the soft robot remains relatively stable for 6 hours.

Fig. S8. Cyclic tests of the DE actuator. (a) Experimental setup. (b) The average maximum and minimum forces during one minute for 6 hours with a 30 mins interval (c) The measured forces in one minute at (i) start, (ii) 2 hours later and (iii) 4 hours later.

Comment 3:

What is the effect of the substrate on the speed of the soft robot? How does paper compare to silicone, glass to sand paper?

Response to comment 3:

According to your comments and suggestion, we have compared the moving speed of the soft robot on various substrates, as shown below and marked in red in the revised SI.

[On Page 14 of the revised SI]

S2.5 Effect of substrates

Results show that the substrate plays a vital role in the moving speed (Movie S14). Five substrates are tested, as shown in Fig. S13. The applied voltage is 6 kV. The moving speed is 7.7, 6.2, 3.4, 3.0, 0.4 mm/s when the substrate is rubber, paper, glass, acrylic and sandpaper, respectively. The friction ratios of the five substrates are around 0.6, 0.5, 0.4, 0.35 and 0.8. When the friction ratio is too small (acrylic and glass), both feet are slippery, and the moving speeds of the soft robots are small. When the friction ratio is too large (sandpaper), the actuation force cannot overcome the friction force, and the soft robot doesn't move. Substrates with suitable friction ratios are required for the soft robot.

Fig. S13. Effects of the substrates. (a) Five different substrates were used: rubber, paper, glass, acrylic and sandpaper. (b) The moving speed of the soft robots on various substrates.

Comment 4:

How does the speed and performance scale with the size of the soft robot? Is it scalable?

Response to comment 4:

Thank you for the comments. The soft robot is scalable. In the revised manuscript, we have designed and fabricated scaled soft robots, and tested their performances. We have designed, fabricated and tested the soft robots with different size and parameters. We have added the following sentences to clarify this point in the revised SI.

[On Page 15 of the revised SI]

S2.6 Scalability

The soft robot is scalable (Fig. S14, Movie S15 and Movie S16). When the soft robot is scaled, the design parameters need to change correspondingly. The critical parameter is the thickness of the frame. When the frame is thick, the saddle shape is too flat to generate enough actuation. When the frame is thin, a saddle shape with large curvatures is formed, which leads to self-contact and causes the soft robot to burn. We have fabricated soft robots with 0.8 and 1.5 original sizes. The original acrylic frame is used for the 0.8-size soft robot, and we can observe that the soft robot can only move forward. Backwards or turning motion cannot be formed as the frame is thick. For the 1.5-size soft robot, we tried two different thicknesses, 0.4 mm and 0.5 mm. When the thin 0.4 mm framework is used, the DEA self-contact and burns under actuation. The soft robot can also move forward when the thick 0.5 mm framework is used. We measured the moving speed of the scaled soft robots. The forward speed of the 0.8-size soft robot is 4.6 mm/s. The forward speed of the 1.5-size soft robot is 4.6 mm/s. Their moving performances are not good, as the thickness of the acrylic frame is not optimal.

Fig. S14. Scalability of the soft robots. Soft robots with (a) 0.8 and (b) 1.5 times the original size.

Comment 5:

Have the authors performed any experiments to incorporate heaters into the design so that the chiral-lattice can be reconfigured without manual input of thermal stimulus?

Response to comment 5:

We thank the reviewer for the comments. We have performed experiments to incorporate heaters into the design (Movie S17). The following paragraph is added in the revised SI to clarify this point.

[On Page 18 of the revised SI]

S3.3 Integrated heating element

We used a ceramic heating plate as the heating element (Fig. S18 and Movie S17). The structure of

the chiral foot is shown in Fig. S18(b). An external DC power with 5V and 1A current is used to heat the feet. Fig. S18(a) shows the moving trajectory of the soft robot with an embedded heater. The soft robot turns right first under a frequency of 45 Hz. We then use external heat to heat the feet. Upon heating, the foot rotates and recovers to its original shape, which forms an angle with the central line of the soft robot. The inclined angle drives the soft robot to turn left.

Fig. S18. Locomotion of a soft robot with embedded heater. (a) The moving trajectory. (b) The structural design of the chiral foot.

Comment 6:

What other applications can the authors envision for the soft robot beyond locomotion? Can it carry a load? What are the effects of a load on the performance? Does a slight change in the load still allow for the lifting of the flat and chiral feet to enable locomotion?

Response to comment 6:

According to your comments and suggestion, we have discussed the potential applications of soft robots and also added new experiments to show the capability of our soft robot to carry payloads. To clarify these points, we have added the following sentences in the revised manuscript and SI.

[On Page 16 of the revised Manuscript]

The developed soft robots have the ability of dexterous motions by controlling the frequency only, which can support a wide range of applications, such as search and rescue, exploration and inspection, etc. The soft robots can navigate through the unstructured environment and squeeze into tight spaces, making them ideal for search and rescue missions. They can navigate complex environments and gather scientific research or inspection data. The electric-driven mechanism also allows for ready integration with sensors and control circuitry, enabling the design of autonomous soft robotics with autonomous behaviors, such as navigation, obstacle avoidance, and task completion, even under external disturbances.

[On Page 16 of the revised SI]

S2.7 Carrying load

We have conducted experiments on the soft robot carrying a load. The soft robot can move forward, backwards or turn right when carrying a 2.2 g screw, which is around 46% of the soft robot's weight (4.8 g) (Fig. S15(a) and Movie S18). When the load increases to 4.1g, the soft robot cannot move.

Fig. S15. The performances of the soft robots carrying loads: (a) a load with 2.2g and (b) a load with 4.1g. The soft robot can move forward and backwards when the load is 2.2g. The soft robot cannot move when the load is 4.1g.

Comment 7:

Minor Comments:

Minor grammar corrections are needed.

Response to comment 7:

We thank the reviewer for the comments. We have carefully modified the grammar errors.

We greatly appreciate your insightful comments and suggestions again, which have substantially improved our manuscript. We hope our revised manuscript and responses can address all of your concerns and comments. Thank you

Response to Reviewer 3

General comments:

This paper proposes a novel strategy to control a soft robot driven by DE. The core module in the robot is the so called chiral-lattice foot. Based on the dynamic response of the foot, the robot can move forward, backward and in circular direction via single voltage input. The innovative design method can inspire the research in soft robot and flexible structures. The operation principle is practical and highly valuable for the robotic system in applications. The manuscript is well organized and clearly presented to accept.

Response to General Comments:

Thank you very much for your time in reviewing our paper and for the insightful and constructive comments. We greatly appreciate your recognition of our work. To address your comments and concerns in full, we have significantly improved our manuscript and added a substantial amount of new data, analyses, discussions, and clarifications (with revised 2 Figure, newly added 11 Supplementary Figures, and revised 2 Supplementary Figures). In the following paragraphs, we will address your comments point-by-point. Sentences newly inserted into the manuscript and the supplementary information are marked in red.

Comment 1:

Comments in details:

More detailed design parameters of the robot should be provided, including both feet and the DEA. Also, the magnitudes of the parameters in Fig. S1 are missed.

Response to comment 1:

Thank you for the comments. In the revised manuscript, we have added the referred information of our robot, which are reflected in the following sentences.

[On Pages 4-7 of the revised SI]

The dimensions of the soft robots are given below. The plain foot is a cuboid with length, width and thickness of 20.00 mm, 20.00mm and 1.00 mm, respectively. The design parameters of the chiral lattice foot are shown in Fig. S2. Both left- and right-handed chiral lattice foot are used in this work. When demonstrating the turning motions, only left-handed chiral lattice foot is used. For the lattice foot, the upper surface of the foot is a square with a side length $D = 15$ mm. The height $H = 2.00$ mm. The lattice foot consists of 8 straight beams. The length $L = 12.45$ mm and diameter $w = 1.5$ mm. All the beams are twisted with $\gamma = 90^\circ$. A triangular prism is located at the bottom of the lattice foot with $\alpha = 7^\circ$, $b = 2.45$ mm. The CAD design of the acrylic frame is shown in Fig. S3. The thickness of the acrylic sheet is 0.3mm. The magnitudes of the parameters in Fig. S1 are $L = 12.45$ mm and $w = 1.5$ mm in Fig. S1(a), $L = 13.95$ mm, $R = 1.975$ mm in Fig. S1(b) and $L = 15$ mm, $w = 1.5$ mm, $R = 2$ mm, $d = 5$ mm in Fig. S1(c).

Fig. S2. The design parameters of the chiral lattice foot. A left-handed chiral lattice foot is shown.

Fig. S3. The CAD design of the acrylic frame. The unit is mm. The thickness of the acrylic sheet is generally 0.3mm.

Comment 2:

The diagram in Fig. 2a should be the front view.

Response to comment 2:

Thank you for the suggestion. We have replotted Fig.2(a) with the front view.

Fig.2(a) A dynamic model is developed to study the bouncing behavior of the chiral-lattice foot.

Comment 3:

In page 7, line 190, "0.0671 N, 0.0941 N and 0.127 N", the accuracy is not uniform. In line 197, the author claimed "The effective elastic constant $k = 38 \text{ N/m}$ based on the finite element simulation results (Fig. S3)". The experimental data of "k" should be provided and compared with the simulation results. And more details of the FEA model should be provided, such as the material property and element type. In addition, the mechanical test data of the materials in this work should be provided. In line 188-200, many design parameters and test data are mentioned, it will be helpful to add more details and illustrations.

Response to comment 3:

We thank the reviewer for the constructive comments and suggestions. We have corrected the numbers to have uniform accuracy. We have conducted experiments to measure the effective elastic constants of the lattice feet. The details of the FEA model and the test data of the materials are provided. To clarify the above issues, we have added the following paragraph in the revised manuscript.

[On Page 7-8 of the revised SI]

To measure the effective elastic constant of the lattice foot, we designed a bilayer lattice foot with symmetric shape, as shown in Fig. S5(a). A compression displacement is applied (Fig. S5(b) and (c)). The strain rate is 1 mm/min. The experimentally and FEM compressive stress and strain relationship of a single lattice foot is shown in Fig. S5(d). The experimental linear fitted k is 35 N/m, 41 N/m and 72 N/m when the strain is 0.05 mm, 0.1 mm and 0.25 mm, respectively. When calculating the critical frequency of the bouncing behavior, the axial deformation of the lattice foot is small. We used an effective elastic constant $k = 38 \text{ N/m}$, which is reasonable. Fig. S5(e) shows the uniaxial tensile stress-strain curve of dog-bone specimens with Vero material. The Young's modulus of material is 1.5GPa which is obtained by linearly fitting the curves with a 3% strain. We choose 3% because the local strain of the material is generally less than 3% under compression.

[On Pages 7-8 of the revised Manuscript]

The angle between the chiral lattice's top surface and the horizontal direction $\theta = 4.8^\circ$. Therefore, the applied force perpendicular to the top surface of the chiral foot can be estimated as $F_z = F_m \times \sin\theta$.

When the applied voltage is 6 kV, the corresponding μ is calculated as 0.79 using $\mu = F_z / 2mg$.

The critical Ω_c is calculated by Eq. (7) as 0.46, 0.47 and 0.50 for $\beta = 0.03, 0.5$ and 1.0, indicating that the effect of β can be neglected. The inclined angle of the designed chiral lattice foot is $\gamma = 90^\circ$.

The effective elastic constant $k = 38 \text{ N/m}$ based on the finite element simulation and experimental results (Fig. S5). The mass of the chiral foot is measured as $m = 0.51 \text{ g}$. The natural frequency of the chiral lattice is then calculated as $f_0 = 2\pi\omega_0 = 43.5 \text{ Hz}$.

The critical frequency is then calculated

as $f_0 \times \Omega_c = \sim 20$ to 23 Hz when β ranges from 0.03 to 1.0.

[On Page 17 of the revised Manuscript]

3-node quadratic hybrid beam elements (B32H) are used with refined meshes to improve computational accuracy. Quasi-static simulations are performed using Abaqus/Standard. The Vero is modeled as a linear elastic material with Young's modulus $E=1.5\text{GPa}$ (Fig. S5(e)).

Fig. S5. The dependence of the compressive force F and axial displacement z . (a) Structural design of a double layer chiral lattice foot. Experimental snapshots of the (b) undeformed and (c) deformed structures. (d) Comparisons of the finite element simulated and experimental compressive force F vs axial displacement Δz of the lattice foot. (e) Uniaxial tensile stress-strain curve of Vero material.

Comment 4:

In Fig. S4e,f, the data under 6 kV input is much fewer than that of 5 kV and 7 kV with frequency ranges from 35 Hz-45 Hz. The reason should be presented.

Response to comment 4:

Thank you for the comments. In the revised manuscript, we have performed more experiments with 6 kV input. The results are added in the revised Fig. S7(e) as follows.

Fig. S7(e). The average maximum forces with different f under 5, 6 and 7 kV.

Comment 5:

In page 8, line 208, the dynamic equations should be Eqs (1) and (2), when the robot bounces off the ground, since z_1 is no longer being zero. In line 224-225, the marker b in Fig. 2b indicates that $\beta=0.5$ in the chiral-lattice system. However, β equals 0.03 in Fig. 2c,d. The parameter should keep the same after calibration.

Response to comment 5:

Thank you for the comments. In the revised manuscript, we have revised the equation labels. We have added the theoretical results using $\beta = 0.5$, as shown in Fig. 2(c) and (d). It can be seen that the dynamic behaviors of the chiral lattice foot will change slightly with the change of β .

Fig. 2. (c) The bouncing diagram of the lattice foot at $\beta=0.5$. The dimensionless flight time ϕ is shown as a function of Ω . (d) Three spatio-temporal diagrams showing the motions of the top (red) and bottom plates (black). The bottom plate stays on the ground at (I) and bounces at II and III.

Comment 6:

In page 9, line 265-266, the authors claimed that, "The dynamic behavior of the lattice foot under a 7kV and 46 Hz voltage is numerically solved.". The model parameters should be presented. In line 269-270, the authors claimed that, "The dependence of the twisting angle α and the axial

displacement Δz is analyzed using the finite element method (Fig. 3(c))." The experimental results should be presented. In line 272, is the chiral-lattice foot left-handed?

Response to comment 6:

We thank the reviewer for the comment. In the revised manuscript, we have added the referred parameters. The model parameters are $L = 0.01$ m, $g = 9.8$ kg/m², $k = 38$ N/m, $m = 0.51$ g, $\eta = 0.07$, $\beta = 0.5$, $F_m = 0.094$ N, $\mu = 0.79$. We have conducted experiments on the dependence of the twisting angle α and the axial displacement Δz . The dependence of Δz on α is shown in Fig. S4(c) and compared with the FEM results. The chiral lattice foot is left-handed.

Fig. 3(c) and S4(c). Comparison between the theoretical and experimental twisting angle α and the axial displacement Δz .

Comment 7:

In page 12, line 330-332, is the "deviation" angle same as γ in Fig. S1? Why different deviation angles lead to different deviation direction of the robot? The relationship between the chirality and deviation angle of the chiral-lattice foot and the deviation direction of the robot should be discussed in detail.

Response to comment 7:

Thank you for pointing it out. We may mention that the deviation angle is not the same as γ . We have presented the mechanism of direction turning as follows, which are reflected in the revised manuscript and SI.

[On Pages 12-14 of the revised SI]

The turning mechanism is illustrated. The turning is controlled by the deviation angle, which is between the triangular prism and the vertical direction. One way to control the turning direction is by manually twisting the chiral lattice foot (Fig.S12(a) and (b)). When the chiral lattice foot is twisted clockwise, the actuation force generated by the prism is perpendicular to the triangular prism

and along the bottom right direction, which drives the soft robot to turn left (Fig. S12(a)). Similarly, the soft robot turns right when the chiral lattice foot is twisted counter-clockwise (Fig. S12(b)). The mechanism of turning right at high frequency is shown in Fig. S12(c). When the chiral lattice foot is compressed, the bottom plate rotates counter-clockwise, which drives the soft robot to turn right.

Fig. S12. The direction of the prism of the chiral lattice foot. The top and front views are shown. (a) The soft robot turns left when the chiral lattice foot is twisted clockwise. (b) The soft robot turns right when the chiral lattice foot is twisted counter-clockwise. (c) When a left-handed chiral lattice foot is compressed, the bottom plate rotates counter-clockwise, and the soft robot turns right.

Comment 8:

In Movie S8, the FEA is presented without details, which should be illustrated in main text or supplementary materials.

Response to comment 8:

We thank the reviewer for the comments. We have added the following FEA in detail in the revised manuscript.

[On Pages 17-18 of the revised Manuscript]

A multi-branch model developed in our previous study ([Advanced Functional Materials, 2023 33.1: 2208849]) is used to capture the thermo-mechanical behaviors of chiral lattice foot. The model consists of one equilibrium branch associated with the elastic response and 30 non-equilibrium

branches associated with the viscoelastic response. The non-equilibrium branches are represented by the Maxwell model. Multi-branch model is achieved by implementing the fitted viscoelastic parameters into the time-domain Prony-series model in ABAQUS material library with a UTRS subroutine to describe the time-temperature superposition. One end of the chiral-lattice foot was fixed, and displacement loading was applied to the other.

We greatly appreciate your insightful comments and suggestions again, which have substantially improved our manuscript. We hope our revised manuscript and responses can address all of your concerns and comments. Thank you

REVIEWERS' COMMENTS

Reviewer #1 (Remarks to the Author):

The reviewer's minor comments on the correlation between the twisting angle and turning motions, as well as the controllability of S-shaped paths are well addressed in the revised version. The reviewer is also very impressed by the new demonstration on the maze navigation to show its controllability in navigation. The reviewer recommends its acceptance for publications as is.

Reviewer #2 (Remarks to the Author):

The new results and improved text by the authors have answered all the questions satisfactorily and this has definitely strengthened the paper. I am happy to recommend the paper for publication in Nature Communications.